# The Gut Microbiome, Aging, and Longevity: A Systematic Review

**DOI:** 10.3390/nu12123759

**Published:** 2020-12-07

**Authors:** Varsha D. Badal, Eleonora D. Vaccariello, Emily R. Murray, Kasey E. Yu, Rob Knight, Dilip V. Jeste, Tanya T. Nguyen

**Affiliations:** 1Department of Psychiatry, University of California San Diego, La Jolla, CA 92093, USA; vbadal@health.ucsd.edu (V.D.B.); evaccariello@health.ucsd.edu (E.D.V.); ermurray@ucsd.edu (E.R.M.); keyu@ucsd.edu (K.E.Y.); djeste@health.ucsd.edu (D.V.J.); 2Sam and Rose Stein Institute for Research on Aging, University of California San Diego, La Jolla, CA 92093, USA; 3Department of Pediatrics, University of California San Diego, La Jolla, CA 92093, USA; robknight@eng.ucsd.edu; 4Department of Computer Science and Engineering, University of California San Diego, La Jolla, CA 92093, USA; 5Department of Bioengineering, University of California San Diego, La Jolla, CA 92093, USA; 6Center for Microbiome Innovation, University of California San Diego, La Jolla, CA 92093, USA; 7Department of Neurosciences, University of California San Diego, La Jolla, CA, 92093, USA; 8VA San Diego Healthcare System, La Jolla, CA 92161, USA

**Keywords:** centenarians, microbes, metabolites, inflammation, immunosenescence, cognition, functional potential, healthy aging

## Abstract

Aging is determined by complex interactions among genetic and environmental factors. Increasing evidence suggests that the gut microbiome lies at the core of many age-associated changes, including immune system dysregulation and susceptibility to diseases. The gut microbiota undergoes extensive changes across the lifespan, and age-related processes may influence the gut microbiota and its related metabolic alterations. The aim of this systematic review was to summarize the current literature on aging-associated alterations in diversity, composition, and functional features of the gut microbiota. We identified 27 empirical human studies of normal and successful aging suitable for inclusion. Alpha diversity of microbial taxa, functional pathways, and metabolites was higher in older adults, particularly among the oldest-old adults, compared to younger individuals. Beta diversity distances significantly differed across various developmental stages and were different even between oldest-old and younger-old adults. Differences in taxonomic composition and functional potential varied across studies, but *Akkermansia* was most consistently reported to be relatively more abundant with aging, whereas *Faecalibacterium*, *Bacteroidaceae*, and *Lachnospiraceae* were relatively reduced. Older adults have reduced pathways related to carbohydrate metabolism and amino acid synthesis; however, oldest-old adults exhibited functional differences that distinguished their microbiota from that of young-old adults, such as greater potential for short-chain fatty acid production and increased butyrate derivatives. Although a definitive interpretation is limited by the cross-sectional design of published reports, we integrated findings of microbial composition and downstream functional pathways and metabolites, offering possible explanations regarding age-related processes.

## 1. Introduction

Aging refers to the process of becoming older, a process that is genetically determined and environmentally modulated [1]. It involves changes in dynamics of biological, environmental, behavioral, and social processes. Primary cellular and molecular hallmarks of aging include genomic instability, telomere attrition, epigenetic alterations, and loss of proteostasis, which lead to compensatory mechanisms such as deregulated nutrient sensing, mitochondrial dysfunction, and cellular senescence; ultimately, these lead to stem cell exhaustion and altered intercellular communication that are responsible for functional decline associated with aging [2]. The rapid development of next-generation sequencing technologies can help unravel the biological and genetic mechanisms of aging and age-related diseases. There is increasing evidence that the gut microbiome lies at the core of many age-associated changes and plays a role in longevity across species [3,4]. Aging has physiological effects on both the host and the microbiome, and host–microbiota interactions may impact aging as a unit [4]. The microbiome is a principal factor in determining the immune system response and its dysregulation may sustain pro-inflammatory states [5]. The progression of aging involves a gradual weakening of the immune system, resulting in an imbalance between pro-inflammatory and anti-inflammatory activity [6]. Age-related changes in pro-inflammatory status result in low-level systemic inflammation (“inflammaging”) that increases the propensity for chronic diseases and disabilities, including cardiovascular disease, cognitive decline, metabolic disease, frailty, and mortality [7,8]. Furthermore, gut microbes can communicate with the brain and modulate behavior, including higher-order cognitive functions, via the “gut–brain axis” through neural, immune, and hormonal mediators [9]. Together, the microbiome offers an exciting perspective to understanding both physical and cognitive aspects of aging.

With a generalized decline in health, it is not clear what “healthy aging” really is. There is no consensus definition of healthy vs. unhealthy aging. A hallmark of aging is heterogeneity. People become more different from one another as they age. Even different tissues in the same body age at different rates. Aging can be studied from different perspectives: “normal” (i.e., average or typical) aging, pathological aging (i.e., associated with specific diseases or other indicators of accelerated aging), and successful aging [10]. The present review focused on normal and successful aging, while excluding disease-related pathological aging. Extremely long-lived individuals, such as centenarians, are examples of highly successful aging. They have avoided or survived most of the diseases that are responsible for morbidity and mortality in most other older adults; however, they may still show some characteristic signs of aging. These oldest-old adults can offer great insights into the most ideal of aging processes. Inflammaging is still present in long-lived people (nonagenarians and centenarians), but less so than “normal” older adults, and their pro-inflammatory status is balanced by concomitant anti-inflammatory responses [11]. Thus, the composition of the gut microbiota throughout the lifespan may modulate health and disease in aging populations.

Unlike numerous published reviews of the microbiome in aging-associated conditions and diseases [12,13,14,15], this investigation focused on aging per se. Additionally, while a number of narrative reviews have been written on the role of the microbiome in aging, including centenarians [16,17,18], no article to our knowledge has systematically reviewed all the available studies of the gut microbiome, including the metabolome, in human normal and successful aging populations. The issue of gut microbiome changes in aging is most aptly addressed through longitudinal studies. However, in our search of the literature, we did not find any longitudinal investigations of the gut microbiome in aging, which is a major limitation of the current literature. Of concern, it is difficult to separate cohort from within-subject effects. Nevertheless, we summarize the current knowledge of the gut microbiome in aging—not only in terms of understanding the composition, function, and metabolic products of the microbiota of older adults and extremely long-lived individuals, but also from the perspective of aging across the lifespan—while acknowledging the limitations of the current literature. Additionally, this study is unique in that we synthesized associations between the gut microbiota and clinical factors, including cognition and living environment, and included studies of interventions targeting the gut microbiome in aging populations.

## 2. Materials and Methods

We performed a search of PubMed, PsycINFO, and Embase for articles published before 13 December 2019 using the following search string: (microbiome OR metabolome) AND (gut OR fecal OR intestinal OR gastrointestinal) AND (“older adults” OR aging OR lifespan) AND (“healthy” OR “no disease”) AND (“humans” OR “clinical population”). See Appendix B for specific search queries. We examined titles and abstracts of all returned citations and reviewed selected full-text articles based upon our inclusion/exclusion criteria.

### 2.1. Inclusion and Exclusion Criteria

Studies were selected if they met the following criteria: (1) utilized high-throughput sequencing methods to quantify microorganisms and/or their functional pathways in the gut or distal large intestine, (2) included a human aging sample, (3) reported results of age analyses (e.g., compared findings between an older group with a younger comparison group; examined an age relationship between the microbiota and other clinical factors within an older adult group), and (4) were published in English. Studies using liquid chromatography coupled with mass spectrometry (LC-MS) analysis to identify gut metabolites were also considered. Aging samples were defined as those that included subjects older than age 65 and were not focused on a specific disease. While the inclusion criteria for participants across individual studies varied, a majority of investigations excluded subjects with major medical co-morbidities (18 out of 27) and recent antibiotic use (15 out of 27). We excluded review papers, meta-analyses, abstracts or conference proceedings, articles with duplicate data, case reports, and studies exclusively using animal models or other non-gut microbiome biomarkers (e.g., saliva, blood).

### 2.2. Review Process

Figure 1 depicts the PRISMA (Preferred Reporting Items for Systematic Reviews and Meta-Analyses) flow chart for inclusion of studies in this systematic review. Our database search yielded a total of 205 articles, after duplicates and non-English reports were removed. The titles and abstracts of the remaining articles were screened based on inclusion/exclusion criteria, and 29 full-text studies were further reviewed for eligibility. Of these, 24 met the above-mentioned criteria, in addition to three articles found through review of the references cited. In total, 27 studies were included in this review.

## 3. Results

### 3.1. Characteristics of Reviewed Studies

We assessed the relevance of gut microbiota composition to the phenomenon of aging and found the studies fell into four categories with broad focus on: (A) the gut microbiota composition of extremely long-lived individuals (e.g., nonagenarians and centenarians), (B) changes and transition in gut microbiota that accompany aging across the lifespan, (C) relationship of the gut microbiota to cognition in older adults, and (D) changes to the gut microbiota following interventions targeting the microbiome in older adults. A summary of relevant data from reviewed studies is presented in Table 1. Detailed sample and methodology characteristics for each study are provided in Appendix A, and detailed results are provided in Appendix A.

(A) Eight out of 27 articles specifically focused on longevity and the gut microbiota in long-living, oldest-old adults. These studies included nonagenarians (90–99 years) and/or centenarians (100+ years) and compared long-living individuals with at least one younger comparison group (e.g., younger adults [20–50 years] or younger-old adults [60–89 years]). (B) Twelve articles investigated changes in the microbiota across the lifespan. This group of articles focused on aging across the lifespan. (C) Three articles investigated the gut microbiota in relation to cognition in older adults. (D) The remaining four articles investigated changes in gut microbiota following treatment or intervention targeting the microbiome in older adults. Methods used to characterize the microbiota varied across studies, with the majority (73%) utilizing 16S rRNA sequencing. Eleven studies investigated the functionality of the microbiota through analysis of genetic functional pathways and metabolomics.

The microbiota can be quantified using different metrics. Below, we report findings for global community diversity, including alpha diversity and beta diversity. A summary of age-related findings for each gut microbiota metric across different categories of studies is presented in Table 2. Alpha diversity is a measure of the within-sample diversity of a community, often described in terms of the number (i.e., richness) or distribution (i.e., evenness) of organisms in a sample. It is commonly observed that low alpha diversity is suggestive of a dysbiotic gut microbiome [46,47,48,49,50]. Beta diversity is a measure of the between-sample differences of pairs of communities. It is worth noting that different measures of alpha and beta diversity may yield different results, and we have detailed the measures used in each study and summarized specific results in Appendix A. Next, we discuss findings of differentially abundant taxa and specific taxonomic compositions associated with aging. Finally, we present data on functional elements, such as genes, associated pathways, and inferred functional potential, and metabolomic signatures.

### 3.2. Alpha Diversity

Thirteen studies reported findings of alpha diversity for microbial taxa.

3.2.A. Long-Lived Individuals

Three of these studies reported higher levels of alpha diversity in the long-living groups, compared to younger adults, including middle-aged and young-old adults [22,24,26]. Two studies did not find any differences in alpha diversity across oldest-old, younger-old, and young adult groups [21,25]. Although Wu and colleagues [25] did not find differences in overall alpha diversity, they found higher diversity of “core microbiota” taxa (i.e., present in at least 50% of samples) in oldest-old adults, compared to young-old and young adults. Three studies reported alpha diversity of functional pathways and metabolites, with two showing increased alpha diversity in young-old adults and oldest-old adults [25,37], whereas one study did not find differences between young-old and oldest-old adults [23]. One study found no differences in alpha diversity between young-old adults with or without major medical illnesses (e.g., diabetes, cancer, cardiovascular, pulmonary, liver, or neurodegenerative diseases) [38].

3.2.B. Lifespan

Across the lifespan, alpha diversity was lowest at infancy, with increasingly higher levels through adolescence and young adulthood (20 years). Diversity levels were stable without differences across adult decades and, then, were higher in young-old adults and oldest-old adults [34]. However, one study did not find differences in alpha diversity between young-old adults and younger adults [42].

3.2.C. Cognition

Lower alpha diversity was associated with poorer cognition, including slower reaction times and worse verbal fluency [41].

3.2.D. Intervention

There were no differences in alpha diversity following probiotic or prebiotic supplementation in older adults [42,44].

### 3.3. Beta Diversity

Thirteen studies analyzed beta diversity.

3.3.A. Long-Lived Individuals

Five studies found significant differences in Bray–Curtis dissimilarity, unweighted UniFrac, and Euclidean distance between oldest-old adults and younger control groups [19,23,24,25,26]. One study reported no difference in beta diversity between centenarians and super-centenarians [19].

3.3.B. Lifespan

Aging explained a significant proportion of variance in unweighted and weighted UniFrac distances [34]. Beta diversity of the gut microbiota was significantly different between community-dwelling individuals and long-term care residents. However, beta diversity between community-dwelling older adults was not different from younger adults [27,30]. The composition of the gut microbiota of long-term residents with stable microbiotas (i.e., low Spearman distance between composition at two time points) was similar to that of community-dwelling individuals, whereas the composition of community-dwelling individuals with unstable microbiotas was more similar to that of long-term care residents [30]. Beta diversity did not differ between young-old adults with or without major medical illnesses [38].

3.3.C. Cognition

Studies did not examine beta diversity.

3.3.D. Intervention

Two studies examined beta diversity, and neither observed differences as a result of probiotic or prebiotic supplementation [42,44].

### 3.4. Taxonomic Composition

Twenty-five studies examined taxonomic composition of the gut microbiota.

3.4.A,B. Integrated Findings for Long-Lived Individuals and Lifespan

Figure 2 depicts taxa that were found to be differentially abundant in nonagenarians and centenarians, compared to younger age groups, across reviewed studies. Six studies reported on phylum level taxonomic differences. Four studies found Proteobacteria to be more abundant in older adults, including centenarians, compared to younger adults [21,25,26,34]. Two studies reported that oldest-old adults had decreased relative abundance of Firmicutes, compared to young-old and younger adults [25,26]. One study found that Bacteroidetes was more abundant with age after age 70 [34]; however, other studies found mixed results among oldest-old and young-old adults [21,26]. Actinobacteria was substantially lower after weaning (i.e., period of transition during infancy that involves a major dietary change from reliance on mothers’ milk) and continued to be lower with age [34]. However, one study reported higher relative abundance of Actinobacteria in oldest-old adults, compared to younger adults [21].

At the family level, *Bacteroidaceae, Lachnospiraceae*, and *Ruminococcaceae* were negatively associated with aging [19], and multiple studies identified *Christensenellaceae* (three articles) and *Synergistaceae* (two articles) to be relatively more abundant in oldest-old adults than younger age groups. With regard to genera, *Eggerthella*, *Akkermansia*, *Anaerotruncus*, and *Bilophila* were positively associated with aging [19], and multiple studies found *Akkermansia* (four articles), *Escherichia* (three articles), *Clostridium* (two articles), *Desulfovibrio* (two articles), *Parabacteroides* (two articles), *Odoribacter* (two articles), *Butyricimonas* (two articles), *Eggerthella* (two articles), and *Anaerotruncus* (two articles) to be relatively higher in oldest-old adults. Conversely, *Faecalibacterium* (six articles), *Prevotella* (two articles), and *Bacteroides* (two articles) were relatively reduced in oldest-old adults. Results were mixed for *Bifidobacterium* [19,20,23] and *Ruminococcus* [21,23,25], with articles reporting conflicting directionality in relative abundances.

At the species level, two studies found *Bifidobacterium longum* to be present across the lifespan [31,35]. Other *Bifidobacterium* species were relatively more abundant in unique age groups: *B. breve* was most prevalent in infants, *B. adolescentis* in adults, and *B. dentium* in older adults [31]. Species involved in decomposing and degrading cellulose were present across the lifespan, whereas some species that produce butyrate (e.g., *Butyricimonas virosa*, *Anaerostipes butyraticus*) were more abundant in younger adults but not nonagenarians [32]. Aging was associated with increased presence of specific *Lactobacillus* species, including *L*. *paracasei*, *L. plantarum*, *L. salivarius*, and *L. delbrueckii*, which were dominant in older adults [32,33,36]. *Clostridia sensu stricto*, *Methanobrevibacter smithii*, and *Bifidobacterium adolescentis* were significantly increased in oldest-old adults, whereas *Faecalibacterium prausnitzii, Dorea longicatena*, *Eubacterium rectale, Bacteroides caccae,* and *Fusobacterium mortiferum* were decreased in oldest-old adults [20,25].

Two studies reported on differences in taxonomic abundance across environments. Older adult residents of rehabilitation hospitals and long-term care facilities exhibited a higher proportion of Bacteroidetes, Proteobacteria, Verrucomicrobia, Actinobacteria, *Parabacteroides*, *Eubacterium*, *Anaerotruncus*, *Lactonifactor*, and *Coprobacillus*, compared to community-dwelling older adults, who showed a higher proportion of Firmicutes, *Coprococcus*, and *Roseburia* [21,27]. Oldest-old adults who were community-dwelling or living in longevity villages had higher relative abundances of *Lactobacillus* than oldest-old adults residing in rehabilitation hospitals and urban environments [21].

3.4.C. Cognition

Three studies examined the relationship of specific bacterial taxa to cognition in older adults. Verrucomicrobia and Firmicutes were positively associated with verbal learning and memory, attention, processing speed, and executive functions [39,40]. Conversely, Bacteroidetes and Proteobacteria were negatively associated with executive function, learning, and memory. Similarly, higher abundances of Burkholderiales and Betaproteobacteria were correlated with slower reaction times [41].

3.4.D. Intervention

*Bifidobacterium* and *Lactobacillus* were the most commonly administered bacteria in studies of probiotic and synbiotic supplementation in older adults [43,44,45]. Older adults who received supplementation of these strains showed resultant increases in *Bifidobacterium*, *Faecalibacterium prausnitzii*, *Lactobacillus* spp., and *Lactobacillus acidophilus*, and a decrease in *Escherichia coli*, compared to the placebo group. With regard to dietary intervention, abundances of *Clostridium* cluster IV and *Bifidobacterium* were not altered following 8 weeks on the RISTOMED optimized diet intervention either alone or in combination with a probiotic supplement [45]. However, subgroup analysis revealed that individuals with low-grade inflammation showed an increase in *Bifidobacterium* following the dietary intervention with adjunctive probiotics. Prebiotic supplementation with pectin did not significantly change gut microbial taxa in either young or older adults [42].

### 3.5. Functional Potential and Metabolites

Twelve studies examined functional pathways and metabolites of the gut microbiota.

3.5.A,B. Integrated Findings for Long-Lived Individuals and Lifespan

Functional pathways related to drug transporters were enriched in older adults, compared to young and middle-aged adults [34], which may be related to more frequent use of medications and antibiotics in these groups. Conversely, older adults had a reduced number of gene families involved in genetic transcription, repair, and defense mechanisms compared to younger adults [35]. Additionally, functional pathways related to genetic information processing were decreased in oldest-old adults, compared to young-old and younger adults; however, oldest-old adults had increased functional pathways related to central energy metabolism (e.g., glycolysis) and respiration [21].

With regard to short-chain fatty acids (SCFA), four studies found decreased functional capacity for butyrate production in older adults, with decreased copies of a butyrate-producing gene (butyryl-CoA:acetate CoA-transferase) [28], lack of presence of butyrate-producing bacteria [32], and reduced abundance of pathways related to carbohydrate degradation and metabolism, which is connected with SCFA production [23,25,35]. Conversely, two investigations reported opposing findings: oldest-old adults had greater functional potential for fermenting SCFA such as propanoate and acetate [25] and higher relative abundances of gamma-aminobutyric acid (GABA) and DL-3-amino isobutyric acid, which are derivatives of butyrate [24], compared to young-old and young adults. Among older adults, individuals who are community-dwelling or in a rehabilitation hospital demonstrated higher gene counts for butyrate, acetate, and propionate production and butyrate- and acetate-producing enzymes, compared to individuals in long-term facilities [27].

Metabolism of aromatic amino acids (tryptophan and phenylalanine) was positively associated with aging [23,25], whereas biosynthesis of amino acids (lysine, isoleucine, tryptophan, and indole) was negatively correlated with age [37] and reduced in oldest-old adults, relative to younger adults and young-old adults [21,25]. Vitamin utilization was altered in oldest-old adults compared to both young-old and younger adults. Oldest-old adults showed decreased vitamin B1 pathways, but increased pathways related to B2 and K2 processing [25].

3.5.C. Cognition

Studies did not examine functional potential or metabolites.

3.5.D. Intervention

Prebiotic supplementation with pectin did not alter fecal metabolite levels of SCFA or branched chain fatty acids [42]. Change in *Bifidobacterium* was positively associated with changes in plasma folate and vitamin B12 concentration among older adults with low-grade inflammation, following intervention of both diet only and diet with a probiotic, but not in those without inflammation [45].

## 4. Discussion

To our knowledge, this is the first systematic review of the gut microbiome and metabolome in human aging and longevity. Our study is unique in that we compiled data from across a wide range of studies: studies of long-lived individuals, cross-sectional lifespan studies, and studies focused on cognition and interventions in older adults. Results of the present review found that alpha diversity is higher with aging among normal and successfully aging older adults. No study reported a negative association of alpha diversity with age. Beta diversity distances were significantly different between older adults and younger adults, even between the oldest-old and younger-old adults. Although differences in taxonomic composition and functional potential varied across studies, *Akkermansia* was most consistently reported to be relatively more abundant with aging, whereas *Faecalibacterium*, *Bacteroidaceae*, and *Lachnospiraceae* were relatively reduced, particularly among oldest-old adults. Older adults have reduced pathways related to carbohydrate metabolism and amino acid synthesis, compared to younger adults. However, oldest-old adults showed some functional differences that distinguished their microbiota from that of young-old adults, such as increased SCFA production and pathways related to central metabolism, cellular respiration, and vitamin synthesis.

The reviewed studies investigating indices of microbial diversity across the lifespan found that beta diversity significantly differs across various developmental stages [34] and continues to diverge even amongst young-old adults and oldest-old adults [19,24,25,26,34]. Alpha diversity was higher in oldest-old adults compared to young-old and younger adults [22,24,26]. Furthermore, oldest-old adults with high alpha diversity exhibited greater temporal stability of microbiota composition over time [30]. Lower alpha diversity was associated with decreased cognition in aging [41]. Moreover, previous studies have shown diminished alpha diversity to be a correlate of metabolic and inflammatory diseases [51,52]. In oldest-old adults and young-old adults, a rich and diverse ecosystem may be indicative of a flexible gut microbiota that is adaptive to perturbations (e.g., illness, medication), and may be a marker of longevity [53].

In adults, Firmicutes largely dominate the gut, followed by Bacteroidetes [34]. Oldest-old adults were found to generally have lower Firmicutes and increased Bacteroidetes abundances, consistent with previous evidence suggesting that the Firmicutes/Bacteroidetes ratio increases in adulthood but declines again in older age [54]. However, the Firmicutes/Bacteroidetes ratio also seemed to be dependent on the residential environment of the oldest-old adults (e.g., community-dwelling vs. hospitalized) [21]. Previous evidence has shown both abnormally elevated and abnormally decreased Firmicutes/Bacteroidetes ratios to be implicated in metabolic and gastrointestinal disorders [55]. Taken together, these results suggest that equilibrium between these core phyla may be indicative of health and longevity, but that this balance might be partially dependent upon unique environmental factors.

The complexity of taxonomic findings associated with aging and numerous microbes reported to be different in older adults may be partially explained by complexity in diet [34] and age-related physiological changes [56]. Taxa that have been previously associated with health and anti-inflammatory activity [57,58,59] were found to be elevated in oldest-old adults, including Verrucomicrobia, *Akkermansia*, *Christensenellaceae, Parabacteroides, Odoribacter, Bifidobacterium,* and *Butyricimonas* [19,21,24,25,26]. *Akkermansia* was the most frequently reported genus in the reviewed studies to be higher in oldest-old adults [19,22,24,38,60]. Previous studies have associated *Christensenellaceae* and *Akkermansia* with metabolic health. *Christensenlellaceae* has been associated with lower body mass index, lower risk of heart disease, and type 2 diabetes [61]. *Akkermansia muciniphila* is one of the few known species of the phylum Verrucomicrobia, and this species is known for its capacity to degrade mucin and promote intestinal integrity by reducing toxicity levels associated with high-fat diets [62,63]. Among the studies investigating cognition and microbiota in older adults, Verrucomicrobia was related to better performance on tasks of psychomotor processing speed, cognitive flexibility, and learning [39,40,41]. Higher abundance of Verrucomicrobia was also associated with improved sleep quality. This evidence suggests that Verrucomicrobia, *Akkermansia*, and *Christensenellaceae* may promote gut homeostasis and healthy aging by reducing adiposity, inflammation, and the later risk for development of metabolic and cognitive dysfunction.

On the other hand, Proteobacteria, which has been previously associated with increased gut inflammation and dysbiosis [64], was more abundant in oldest-old adults than in young-old or younger adults [21,25,26]. Additionally, *Faecalibacterium*, which has an important role in the production of the SCFA butyrate [65], was less abundant in oldest-old adults [19,21,25]. At first glance, these patterns appear to be conflicting and counterintuitive to the picture of longevity. However, they also suggest that the gut ecosystem of the oldest-old comprises a delicate balance between health-promoting vs. health-degrading bacteria. It has been previously noted that exceptionally long-living individuals exhibit a complex balance of pro- and anti-inflammatory features, permitting an effective immune response that is counterbalanced by robust anti-inflammatory activity [66,67,68]. Thus, successful interplay between opposing immune response networks may permit oldest-old adults to evade typical age-related pathology.

There are several limitations to this review. Important among them is the cross-sectional design of most of the studies reviewed, which makes it difficult to separate cohort effects from within-subject effects and determine whether these oldest-old individuals exhibited a distinct gut microbiota throughout their lifespan or whether there is a distinct microbiota composition unique to this stage of life. Due to this, it is impossible to draw definitive conclusions about the longitudinal trajectories of bacterial counts or alpha diversity across the lifespan. Additionally, while a majority of studies excluded subjects with major medical co-morbidities or recent antibiotic use, there was variability in inclusion criteria across studies with regard to the specific health status of participants, which may contribute to heterogeneity in findings across different studies. It is also unclear to what extent these findings are genetically vs. environmentally driven. Longevity likely reflects a combination of these factors. Results from previous genome-wide association studies suggest that a combination of gene and environment interactions plays a role in shaping the gut microbiota [69,70]. For some, but not all, microbial taxa, host genetics may actually be a stronger predictor than environmental factors. In a twin-study, *Chistensenellaceae* had the most robust association with host genetics [71]. A similar association between the abundance of *Akkermansia*, *Odoribacter,* and *Bifidobacterium* and host genetics has also been identified [72,73,74]. Thus, the gut microbiota of healthy aging may be partially influenced by host genetics.

Environmentally, the composition and function of the gut microbiota is strongly influenced by both short-term and long-term dietary habits throughout the lifespan [75,76]. With aging, decreases in appetite, loss of teeth, decrease in gustatory perception, and decreased efficiency of the digestive system reduce absorption of essential nutrients, which may influence the microbiota and subsequently health [27,77]. We found that older adults, including the oldest-old adults, have reduced pathways related to carbohydrate metabolism and amino acid synthesis [23,25,37]. However, oldest-old adults showed some functional differences distinguishing their microbiota from that of young-old adults, such as increased SCFA production and pathways related to central metabolism, cellular respiration, and vitamin B2 and K2 synthesis [21,25]. Interestingly, two studies noted that their samples of oldest-old adults did not report the typical age-related changes in appetite and reported regularly eating full meals [21,25]. Maintaining a balanced diet in older age may be a key factor in promoting longevity. Indeed, a diet rich in micronutrients and low in saturated fats has been identified as a common denominator among countries with the highest life expectancies [78]. Foods consisting of resistant starches, for example, would likely promote SCFA production and decrease gut inflammation [79]. Vitamin B1 and K2, which are derived from fermentable foods, lean meats, and whole grains, are of particular importance for host immunity, promoting bone health, and reducing the risk of heart disease [80,81].

A few studies acknowledged the possible role of sex [34] and decline in physical activity levels with age [28] on the microbiome and its impact on cognition [41]. Only one examined beta diversity clustering by sex and found no differences in UniFrac distances between males and females from infants to centenarians [34]. While sex differences in the gut microbial composition have been documented [82], the influence of sex on the gut microbiome may be less influential than other clinical factors, such as genetics [83] or geographical origin [84]. As a result, it is unclear to what extent sex differences in the microbiome or physical activity might explain sex differences in longevity [85]. Physical activity is another important environmental factor that may influence the gut microbiome as well as aging and longevity. One study found that frailty, of which physical activity is a component, moderated the relationship between the microbiota and cognition [41], but no other article examined the role of exercise on the microbiome in aging, thus limiting conclusions that can be drawn.

The dynamic and modifiable nature of the gut microbiome presents exciting opportunities for therapeutic interventions to address health challenges related to aging. Probiotics (live microorganisms) and prebiotics (nondigestible food components that are selectively fermented by intestinal bacteria) have been shown to confer benefits for a variety of health conditions [86]. This review identified several studies investigating changes in the gut diversity and composition following treatment or intervention targeting the microbiome. None of the studies reported significant differences in alpha or beta diversity following probiotic, prebiotic, or synbiotic (probiotic–prebiotic combination) treatments in older adults. Despite a lack of broad-scale compositional changes, supplementation did lead to increases in health-promoting lactic acid bacteria (e.g., *Bifidobacterium*, *Faecalibacterium prausnitzii*, *Lactobacillus* spp.). One study investigated the effect of treatments on SCFA metabolites, and did not observe any significant changes following prebiotic supplementation. It is possible that combining prebiotics and probiotics [43] or diet and probiotics [45] may bring about more robust changes than either treatment alone [42,45]. One study investigated the gut microbial effects of a dietary intervention with adjunctive probiotic treatment and found increased levels of *Bifidobacterium* among individuals with higher levels of systemic inflammation [45]. It is worth noting that the Mediterranean diet (e.g., high consumption of vegetables, legumes, fruits, nuts, olive oil, and fish; low consumption of red meat, dairy products, saturated fats, and processed foods) has been associated with improved health status, including reduced risk of mortality and occurrence of diseases of aging such as cardiovascular disease, cancer, and neurodegenerative disorders [87]. Prior studies have shown that intervention of and adherence to the Mediterranean diet is associated with lower Firmicutes–Bacteroidetes ratio, increased abundances of *Christensenellaceae* and *Faecalibacterium prausnitzii*, increased gene richness (particularly in those with low inflammatory status), and higher levels of gut SCFA in the general adult population [88,89]. Future studies should investigate the effects of the Mediterranean diet in older adults.

## 5. Conclusions

Overall, these findings suggest that longevity may be characterized by increased flexibility and stability of the gut microbiota. Moreover, a particular hallmark of successful aging may be a balance amongst core microbiota as well as a balance between pro- and anti-inflammatory activity. Indeed, what might make the oldest-old adults unique is the ability to maintain (or possibly upregulate) anti-inflammatory activity despite a concomitant uptick of pro-inflammatory activity that occurs in all older adults. The hypothesis is supported by the observed increases in health-promoting taxa and SCFA production. Lastly, a combination of genetic and environmental factors, such as dietary habits, may shape the gut microbiota of older adults. Future prospective longitudinal studies are needed to understand causal relationships between the gut microbiome and aging and longevity. Given the variability among samples and power needed for high-throughput sequencing analyses, investigations involving larger sample sizes are needed. Larger samples will also allow for more complex models that can account for important demographic, lifestyle, and biological factors that might impact microbial composition and health in older adults. Finally, there is an important need for closer collaborations among researchers in the microbiome and aging fields to design innovative studies that leverage knowledge in cutting-edge analytical and computational technologies, in addition to clinical expertise, in understanding the nuances of aging.

## Figures and Tables

**Figure 1 nutrients-12-03759-f001:**
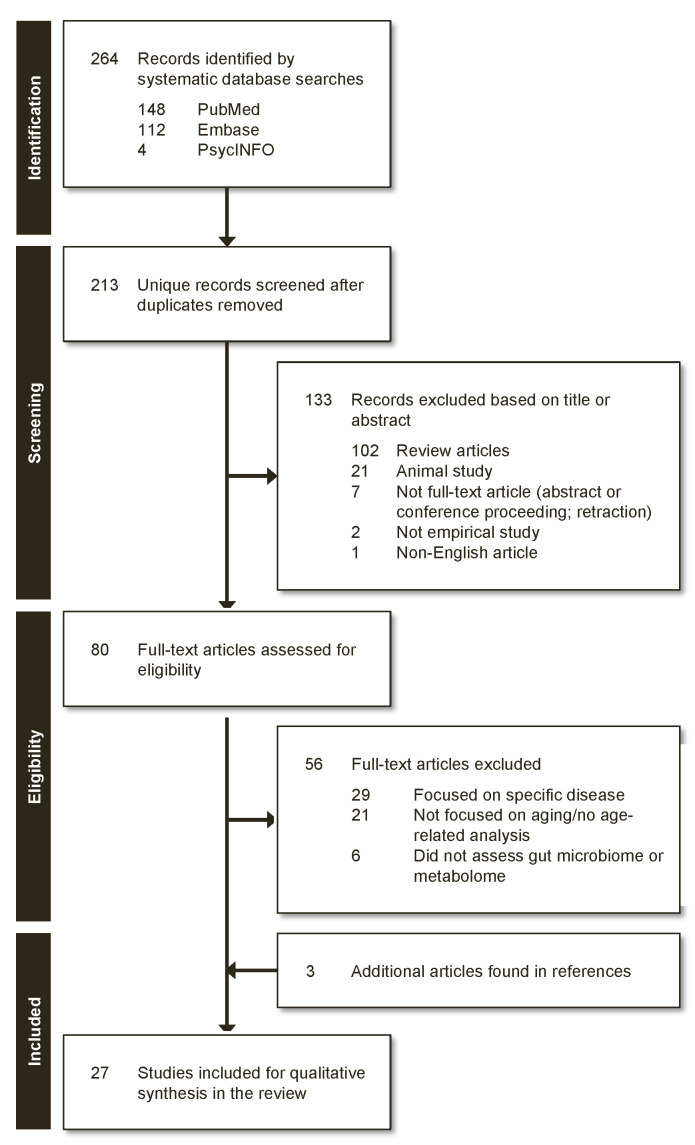
PRISMA-based selection flow chart of reviewed articles.

**Figure 2 nutrients-12-03759-f002:**
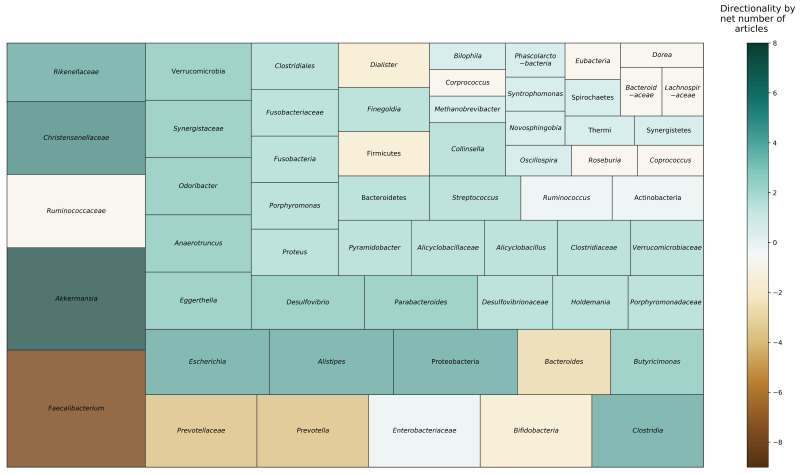
Microbial taxa reported to be differentially abundant in long-living individuals across reviewed studies. Area represents proportion of studies reporting each taxon. Color indicates the direction and net strength of observed findings. Data in this figure are based on nine studies [19,20,21,22,23,24,25,26,34].

**Table 1 nutrients-12-03759-t001:** Summary of reviewed studies.

Publication	Country (Rural/Urban)	Sample Size (Age Range)	Gender (%Female)	Sequencing/Genetic Analysis	Diversity Patterns	Taxonomic Composition	Functional Potential and Metabolites
*Centenarian Studies*							
Biagi et al., 2016 [19]	Italy (U)	SCTN: 24 (105–109) LL: 15 (99–104) YO: 15 (65–75) ADT: 15 (22–48)	SCTN: 75.0 LL: 93.3 YO: 46.7 ADT: 53.3	16S rRNA (V3-V4)	α: N/A β: Different between all possible comparisons of age groups (SCTN, LL, YO, ADT), except between SCTN and LL (unweighted UniFrac)	Changes with aging (SCTN, LL, YO, ADT): Family: ↓ *Bacteroidaceae*, ↓ *Lachnospiraceae*, ↓ *Ruminococcaceae*, ↑ Synergistaceae, and ↑ Christensenellaceae with aging	N/A
Drago et al., 2012 [20]	Italy (U)	CTN: 14 (100–104) ADT: 10 (24–57)	CTN: not reported ADT: not reported	16S rDNA pyrosequencing	α: N/A β: N/A	Family/Genera: CTN ↓ *Enterobacteriaceae*, ↓ *Bifidobacteria*, ↓ *Bacteroides*, compared to ADT Species: CTN ↑ *Bifidobacterium longum*, ↑ *Clostridia sensu stricto*, compared to ADT	N/A
Kim et al., 2019 [21]	South Korea (R)	LL: 30 (95–108) YO: 17 (67–69) ADT: 9 (26–43)	LL: 90.0 YO: 41.2 ADT: 33.3	16S rRNA (V1-V3) pyrosequencing	α: No difference across LL, YO, and ADT groups (Shannon, observed OTUs) β: Did not report across LL, YO, and ADT	Phylum: LL ↑ Verrucomicrobia, compared to YO, and ↑ Verrucomicrobia, ↑ Proteobacteria, ↑ Actinobacteria, compared to ADT; YO ↓ Bacteroidetes and ↑ Proteobacteria, compared to ADT Family/Genera: LL ↓ *Faecalibacterium*, ↓ *Prevotella*, ↑ *Escherichia*, ↑ *Akkermansia*, ↑ *Clostridium*, ↑ *Collinsella*, ↑ *Streptococcus*, ↑ uncultured *Christensenellaceae*, compared to YO and ADT	KEGG Level 1: LL and ADT ↑ pathways related to metabolism, compared to YO; ↓ pathways related to genetic information processing in LL, then YO, then ADT; LL and YO ↑ pathways related to environmental information processing, compared to ADT. KEGG Level 3: 26 metabolic pathways different between groups; of these, LL ↑ phosphatidylinositol signaling system, compared to YO and ADT; LL and ADT ↑ glycosphingolipid biosynthesis, compared to YO, and LL ↑ N-glycan biosynthesis, compared to YO and ADT
Kong et al., 2016 [22]	China (U)	LL: 67 (90–102) ADT: 101 (24–83)	LL: 61.2 ADT: 45.8	16S rRNA (V4-V5)	α: LL ↑ observed OTUs, ↑ Chao, ↑ Shannon, compared to ADT β: Did not assess between LL vs. ADT	Family/Genera: LL ↑ *Ruminococcaceae*, ↑ *Christensenellaceae*, ↑ *Clostridium* cluster XIVa, ↑ *Akkermansia*, compared to ADT	N/A
Rampelli et al., 2013 [23]	Italy (U)	LL: 3 (99–102) YO: 5 (59–75) ADT: 1 (38)	LL: not reported YO: not reported ADT: not reported	Shotgun metagenomic sequencing	α: N/A β: Different between LL and YO (Euclidean distance)	Genera: LL ↑ *Escherichia*, and ↑ *Ruminococcus*, compared to YO; YO ↑ *Faecalibacterium*, ↑ *Eubacterium*, and ↑ *Bifidobacterium*, compared to LL	α: No differences between LL and YO for KEGG pathways (Simpson index) β: LL different from YO and ADT for KEGG pathways (Euclidean distance) Aging (LL, YO) associated with ↑ metabolism of aromatic amino acids (tryptophan and phenylalanine), metabolism of amino acids (tyrosine, valine and lysine); ADT profile associated with ↑ metabolism amino acids (histidine) and carbohydrates (glucose, galactose), pyruvate, and butanoate, and ↑ SCFA production.
Tuikhar et al., 2019 [24]	India (R)	LL: 30 (97–110) ADT internal: 30 (28–47)ADT external: 30 (22–50)	LL: 50.0 ADT internal: 60.0ADT external: 50.0	16S rRNA (V4-V5) qPCR	α: LL ↑ Chao1, compared to ADT; no difference in Shannon index β: Different between LL and ADT (Bray–Curtis)	Family/Genera: LL ↓ *Prevotellaceae*, ↑ *Eggerthella*, ↑ *Rikenellaceae*, ↑ *Alistipes*, ↑ *Porphyromonadaceae*, ↑ *Parabacteroides*, ↑ *Porphyromonas*, ↑ *Odoribacter*, ↑ *Butyricimonas*, ↑ *Alicyclobacillaceae*, ↑ *Alicyclobacillus*, ↑ *Clostridiaceae_Finegoldia*, ↓ *Ruminococcaceae*, ↓ *Faecalibacterium*, ↑ *Anaerotruncus*, ↑ *Enterobacteriaceae*, ↑ *Desulfovibrionaceae*, ↑ *Desulfovibrio*, ↑ *Synergistaceae*, ↑ *Pyramidobacter*, ↑ *Verrucomicrobiaceae, ↑* *Akkermansia* and ↑ *Clostridiales Family XI Incertae Sedis*, compared to ADT Species: LL ↑ *Alistipes shahii*, ↑ *Porphyromonas uenonis*, ↑ *Odoribacter splanchnicus*, ↑ *Parabacteroides goldsteinii*, ↑ *Alicyclobacillus acidoterrestris*, ↑ *Finegoldia magna*, ↑ *Clostridium aminobutyricum*, ↑ *Clostridium p_enrichment_culture_clone_7_25, ↑ Clostridium sp_Kas107_1*, ↑ *Clostridium hathewayi*, ↑ *Eubacterium siraeum*, ↑ *Clostridium cellulolyticum*, ↑ *Clostridium asparagiforme*, ↑ *Faecalibacterium prausnitzii*, ↑ *Clostridium methylpentosum*, ↑ *Anaerotruncus colihominis*, ↑ *Escherichia albertii*, ↑ *Pyramidobacter piscolens*, ↑ *Akkermansia muciniphila,* compared to ADT	109 out of 871 metabolites significantly different between LL and ADT. LL ↑ DL-3-Aminoisobutyric acid, ↑ N-Ethylglycine, ↑ gamma-Aminobutyric acid (GABA), ↑ Imidazoleacetic acid, ↑ Niridazole, ↑ Erucic acid, ↑ Dihydroxyphthalic acid, ↑ Nitridazole, ↑ Triacetin, ↑ Goralatide, compared to ADT internal and external; ↓ cyclohexanecarboxylic acid, compared to ADT internal; ↓ 13-cis,16-cis-Docosadienoic acid, compared to ADT external
Wu et al., 2019 [25]	Italy (R)	LL: 19 (99–107) YO: 23 (68–88) ADT: 17 (21–33)	LL: 68.4 YO: 56.5 ADT: 58.8	Shotgun metagenomic sequencing	α: No difference across age groups (LL, YO, ADT) (Shannon index, observed OTUs) β: LL different from YO and ADT (Bray–Curtis)	Phyla: LL ↑ Proteobacteria, compared to YO and ADT; LL ↓ Firmicutes and ↓ Firmicutes/Bacteroidetes ratio, compared to YO Genera: LL ↓ *Faecalibacterium,* ↓ *Ruminococcus*, ↓ *Corprococcus*, ↓ *Dorea*, ↑ *Methanobrevibacter*, compared to YO and ADT Species: LL ↓ *Faecalibacterium prausnitzi*, ↓ *Eubacterium rectale*, ↑ *Bifidobacterium adolescentis*, ↓ *Ruminococcus sp_5_1_39BFAA*, ↓ *Dorea longicatena*, ↑ *Methanobrevibacter smithii*, compared to YO and ADT	α: LL ↑ Shannon and ↑ observed KOs, compared to YO and ADT; no difference between YO and ADT β: LL different from YO and ADT (Bray–Curtis); no difference between YO and ADT 115 out of 463 gene pathways significantly different among age groups LL ↑ pathways related to central metabolism (glycolysis, pentose phosphate pathways, and tricarboxylic acid cycle), ↑ anaerobic respiration, ↑ aerobic respiration, ↑ metabolism of and fermentation to SCFAs (propanoate and acetate), ↓ amino acid biosynthesis pathways (e.g., L-lysine-, L-isoleucine-, and L-methionine), ↑ aromatic compounds (e.g., L-phenylalanine metabolism and chorismite biosynthesis), ↓ pathways related to carbohydrate degradation, ↑ vitamin B2 and K2 synthesis pathways, ↑ KOs related to phosphotransferase system, F420, and coenzyme M, compared to YO and ADT; LL and YO ↓ vitamin B1 synthesis pathways, compared to ADT
Yu et al., 2015 [26]	China (R)	LL: 21 (50–95) CK: 28 (range not reported; mean: 50)	LL: 52.4 CK: Not reported	16S rRNA (V4) qPCR	α: LL ↑ Chao1 and Shannon index, compared to CK β: Different between CK and LL (unweighted UniFrac)	Phylum: LL ↓ Firmicutes, ↑ Bacteroidetes, ↑ Proteobacteria, ↑ Verrucomicrobia, ↑ Spirochaetes, ↑ Synergistetes, ↑ Thermi, compared to CK Genera: LL ↑ *Escherichia*, ↑ *Phascolarctobacterium*, ↑ *Parabacteroides*, ↑ *Desulfovibrio*, ↑ *Syntrophomonas*, ↑ *Novosphingobium*, ↓ *Faecalibacterium*, compared to CK	N/A
*Lifespan Studies*							
Claesson et al., 2012 [27]	Ireland (U)	YO: 178 (64–102) ADT: 13 (28–46)	YO: not reported ADT: not reported	16S rRNA (V4)	α: Did not report between YO and ADT β: No difference between community-dwelling YO and ADT	Genus: YO ↓ *Ruminococcus*, ↓ *Blautia,* ↑ *Escherichia*/*Shigella,* compared to ADT	Did not report between YO and ADT
Hippe et al., 2011 [28]	Austria (U)	YO: 15 (range not reported; mean: 86) ADT vegetarians: 15 (range not reported; mean: 26) ADT omnivores: 17 (range not reported; mean: 24)	YO: not reported ADT vegetarians: not reported ADT omnivores: not reported	16S rRNA genes and metabolic genes qPCR	α: N/A β: N/A	Genus: YO ↓ *Clostridium cluster XIVa*, compared to ADT omnivores and ADT vegetarians Species: (Melt curve analysis) YO ↓ *Eubacterium hallii*/*Anaerostipes coli*, ↓ *E. rectale*/*Roseburia* spp., ↓ *F. prausnitzii* melt peaks, compared to ADT omnivores and ADT vegetarians	YO ↓ butyryl-CoA:acetate CoA-transferase gene, compared to ADT; ↑ ADT vegetarians, compared ADT omnivores.
Hopkins et al., 2002 [29]	United Kingdom (U)	NHYO: 4 (68–73) YO: 5 (67–88) ADT: 7 (21–34) CHD: 10 (16 months-7)	NHYO: not reported YO: not reported ADT: not reported CHD: not reported	16S rRNA	α: N/A β: N/A	Genus: NHYO ↓ *Bacteroides*, compared to CHD, ADT, and YO; NHYO ↑ *lactobacillus*, ↑ *clostridia*, compared to ADT and YO; YO, NHYO ↓ *Bifidobacteria*, compared to CHD, ADT. Family: ADT ↓ *Enterobacteria*, compared to CHD, NHYO	NHYO ↑ Saturated straight chain (20:0), ↑ Unsaturated straight chain (20:1 cis ll), ↓ Saturated straight chain (12:0, 15:0) and absence of the branched chain (15:O ante and 15:O iso fatty acids); ADT ↑ branched chain CFA, compared to all other groups; ↑ dimethyl acyl (18.1 cisl1 DMA, 14.0 DMA), compared to CHD and NHYO; ↑ 15:0 ante DMA, compared to other groups; CHD did not have dimethyl acyl (18:0 DMA), unsaturated straight chain (16:1 cis9), compared to other groups.
Jeffery et al., 2016 [30]	Ireland (U)	YO: 371 (64–102) ADT: 13 (28–46)	YO: not reported ADT: not reported	16S rRNA (V4)	α: Did not report between YO and ADT β: No difference between ADT and community-dwelling YO (unweighted UniFrac)	N/A	N/A
Kato et al., 2017 [31]	Japan (U)	Age Groups 100: 5 (100 and up) 90: 19 (90–99) 80: 51 (80–89) 70: 31 (70–79) 60: 42 (60–69) 50: 29 (50–59) 40: 37 (40–49) 30: 114 (30–39) 20: 42 (20–29) 10: 10 (10–19) 4: 17 (4–9) 3: 21 (weaned—3 years old) 2: 12 (weaning; mean: 0.8) 1: 16 (preweaning; mean: 0.3)	100: 100.0 90: 78.9 80: 66.6 70: 61.3 60: 66.7 50: 55.2 40: 64.9 30: 52.6 20: 61.9 10: 30.0 4: 58.8 3: 52.4 2: 41.7 1: 43.8	16S rDNA qPCR	α: N/A β: N/A	Species: *B. longum* detected in all groups; Elderly ↑ *B. dentium, ↓ B. catenulatum*; Adult ↑ *B. adolescentis*, ↓ *B. breve*, ↑ *B. gallinarum*, ↑ *B. catenulatum*; Infant ↑ *B. breve*, ↓ *B. adolescentis*	N/A
Kushugulova et al., 2015 [32]	Kazakhstan (not reported)	LL: 6 (90 and up) YO: 17 (50–70)ADT: 6 (30–44)	LL: 100.0 YO: 100.0 ADT: 100.0	16S rDNA	α: N/A β: N/A	Phylum: ADT ↑ Bacteroidetes; YO ↑ Firmicutes; LL ↑ Tenericutes, compared to other groups Species: LL ↓ butyrate-producing and mucin-degrading species, compared with YO, ADT	N/A
Le Roy et al., 2015 [33]	Estonia (U)	YO: 33 (65–81) ADT: 16 (20–48)	YO: not reported ADT: not reported	16S 23S rRNA intergenic spacer region qPCR	α: N/A β: N/A	YO ↑ *L. paracasei*, ↑ *L. plantarum*, ↓ *L. salivarius*, and ↓ *L. helveticus*, compared to ADT	No difference between ADT and YO in metabolic profiles
Odamaki et al., 2016 [34]	Japan (not reported)	Age Groups 100: 6 (100 and up) 90: 19 (90–99) 80: 48 (80–89) 70: 15 (70–79) 60: 28 (60–69) 50: 25 (50–59) 40: 34 (40–49) 30: 88 (30–39) 20: 40 (20–29) 10: 10 (10–19) 4: 14 (4–9) 3: 18 (weaned—3 years old) 2: 12 (weaning; mean: 0.8) 1: 14 (preweaning; mean: 0.3)	100: 100.0 90: 78.9 80: 66.7 70: 66.7 60: 60.7 50: 52.0 40: 61.8 30: 48.9 20: 60.0 10: 30.0 4: 57.1 3: 44.4 2: 50.0 1: 50.0	16S rRNA (V3-V4) qPCR	α: ↑ with age (Chao1, number of observed species, Shannon index, phylogenetic distance whole tree) β: Variation in data due to age (UniFrac distances, both weighted and un-weighted analyses)	Composition across all ages (0 to 100+): Phyla: With ↑ Age, ↓ Actinobacteria, ↑ Bacteroidetes, ↑ Proteobacteria	Infant/Elderly vs. Adult enriched clusters: Preweaned infants ↓ xylose transporterInfant/Elderly ↑ drug transporters
Odamaki et al., 2018 [35]	Japan (U)	Age Groups 100: 6 (100 and up) 90: 19 (90–99) 80: 51 (80–89) 70: 31 (70–79) 60: 42 (60–69) 50: 34 (50–59) 40: 37 (40–49) 30: 117 (30–39) 20: 42 (20–29) 10: 10 (10–19) 4: 17 (4–9) 3: 22 (weaned—3 years old) 2: 12 (weaning; mean: 0.8) 1: 13 (preweaning; mean: 0.3)	100: 100.0 90: 78.9 80: 66.7 70: 61.3 60: 66.7 50: 58.8 40: 62.2 30: 52.1 20: 66.7 10: 30.0 4: 58.8 3: 50.0 2: 33.3 1: 46.2	16S rRNA (V3-V4) Strain-specific PCR	α: N/A β: N/A	Across age groups (preweaning to 100+ age) Species: *Blautia wexlerae, Streptococcus salivarius, Bifidobacterium longum*; *** no inferential statistics provided, detected >50% of participants across age groups	Younger vs. Older (GF enriched in *B. longum* subsp. *longum* strains)Older (GF:11) ↓ GF involved in carbohydrate transport and metabolism, compared to infants (GF:22); Adults ↑ GF involved defense mechanisms, transcription and replication, recombination, and repair, compared other groups. 169 GF enriched in *B. Longum* subsp. *longum* strains in younger participants vs. 55 GF enriched in older participants; younger participants ↑ sialidase-encoding cluster, ↑ an α arabinofuranosidase gene cluster, ↑ pNAC3 (a 10 kb plasmid) homologue, ↑ capsule biosynthesis-related genes and a Type VII secretion system, ↑ some prophage regions found in the AH1206 episome; infants enriched in sialidase clusters; older ↑ extracellular α-L-arabinofuranosidases, putative multidrug-family ABC transporter (associated two-component system), a genetic cluster (Hsp20-family heat shock chaperone), ↑ prophage regions
Pan et al., 2016 [36]	China (R)	LL Bama: 8 (80–99) LL Nanning: 8 (80–99)	LL Bama: 62.5 LL Nanning: 50.0	16S rRNA (V2-V3) PCR-DGGE	α: No difference between LL Nanning and LL Bama subjects (only for diversity of genus *Lactobacillus*; Shannon–Wiener) β: N/A	Representative *Lactobacillus* species in LL: *W. confusa*, *L. mucosae*, *L. crispatus*, *L. salivarius*, and *L. delbrueckii*	N/A
Ruiz-Ruiz et al., 2019 [37]	Spain (U)	YO: 10 (68–81) ADT: 10 (27–44) CHD: 10 (2–5)	YO: 70.0 ADT: 50.0 CHD: 50.0	LC-MS	α: N/A β: N/A	N/A	α: YO ↑ compared to CHD, ADT (microbial richness, Pielou’s evenness, Shannon index) YO ↓ tryptophan and indole production with ↑ age; YO ↓ TnaA, ↓ TrpB, ↓ tryptophan, ↓ indole, compared to CHD, ADT
Singh et al., 2019 [38]	USA (R)	YO: 33 (70–82) NHYO: 32 (70–82)	YO: 57.6 NHYO: 46.9	16S rRNA (V1-V3)	α: No significant differences between YO and NHYO (Shannon, Chao1) β: No difference between YO and NHYO (Bray–Curtis)	Family/Genus: YO ↑ *Akkermansia*, ↑ *Erysipelotrichaceae UCG-003*, ↑ *Bacteroides*, ↓ *Streptococcus*, ↓ *Lactobacillus*, ↑ *Lachnospiraceae* (UCG-005)^1^, ↓ *Escherichia*^1^/*Shigella*^1^, ↑ *Cardiobacterium*, ↑ *Neisseria*, ↑ *Comamonas*, ↑ *Capnocytophaga*, ↓ *Bifidobacterium*, ↑ *Filifactor*, ↑ *Fusobacterium*, ↑ *Propionibacterium*, ↑ *Haemophilus*, ↑ *Corynebacterium*, ↓ *Rothia*, ↑ *Porphyromonas*, ↑ *Ruminococcaceae* UCG-014, ↑ *Prevotella* 2, ↑ *Peptoclostridium*, compared to NHYO	N/A
*Cognition Studies*							
Anderson et al., 2017 [39]	USA (not reported)	YO: 37 (50–85)	YO: 73.0	16S rRNA	α: N/A β: N/A	Verrucomicrobia and Lentisphaerae: ↑ sleep quality Verrucomicrobia: ↑ word reading processing speed Lentisphaerae: ↑ cognitive flexibility; non-significant after accounting for sleep	N/A
Manderino et al., 2017 [40]	USA (U)	YO: 25 (50–85) NHYO: 18 (50–85)	YO: 32.0 NHYO: 33.3	16S rRNA	α: N/A β: N/A	YO ↓ Bacteroidetes, ↓ Proteobacteria, ↑ Firmicutes, ↑ Verrucomicrobia, compared to NHYO Phylum: ↑ Verrucomicrobia showed ↑ verbal learning, ↑ visual scanning, ↑ cognitive set-shifting, ↑ cognitive flexibility (word reading), ↑ cognitive flexibility (color naming); ↑ Firmicutes showed ↑ spatial perception and visual memory, ↑ memory; ↑ Bacteroidetes correlated to ↓ spatial perception and visual memory, ↓ memory; ↑ Proteobacteria correlated to ↓ verbal Recognition/Discrimination, ↓ FAB, ↓ FAS	N/A
Verdi et al., 2018 [41]	United Kingdom (U/R)	YO: 1551 (40–89)	YO: 66.8	16S rRNA	α: ↑ Chao1, phylogenetic diversity, and observed OTU associated with ↓ reaction time and ↓ verbal fluency β: N/A	Order: ↑ Burkholderiales associated with ↓ reaction time Class: ↑ Betaproteobacteria associated with ↓ reaction time	N/A
*Intervention Studies*							
An et al., 2019 [42]	Netherlands (U)	YO pectin: 24 (65–75) YO placebo: 24 (65–75) ADT pectin: 25 (18–40) ADT placebo: 27 (18–40)	YO pectin: 37.5 YO placebo: 50.0 ADT pectin: 68.0 ADT placebo: 48.1	16S rDNA (V5-V6)	α: No difference (Faith’s PD, inverse Simpson) in either ADT and YO, before vs. after pectin supplementation β: Smaller intra-individual change, compared to inter-individual change (weighted UniFrac and unweighted UniFrac), before vs. after pectin supplementation	YO ↑ *Enterorhabdus*, ↑ *Ruminiclostridium* 6, ↑ uncultured genus within the family Coriobacteriaceae, ↑ *Mogibacterium*, ↑ *Lachnospiraceae* (UCG-008), compared to ADT YO ↑ *Enterorhabdus*, ↑ uncultured genus within the family *Coriobacteriaceae*, ↑ *Mogibacterium*, ↑ *Lachnospiraceae* UCG-008), compared to ADT, after pectin supplementation	No significant differences in BCFA and SCFA between YO and ADT at baseline. No significant differences in BCFA and SCFA (acetic acid, propionic acid, butyric acid, valeric acid, isobutyric acid, isovaleric acid) in YO or ADT, before vs. after pectin supplementation
Björklund et al., 2011 [43]	Finland (U)	YO synbiotic: 23 (above 65) YO placebo: 24 (above 65)	YO synbiotic: 79.2 YO placebo: 69.6	qPCR, non-selective DNA-based method, percent guanine-plus cytosine (%G+C) profiling	α: N/A β: N/A	Genera: Synbiotic ↑ *Bifidobacteria*, ↑ *L. acidophilus* *NCFM*, compared to placebo; both (synbiotic and placebo) ↓ *Clostridium* cluster XIVab, ↓ *Blautia coccoides*– *Eubacterium rectale*	N/A
Spaiser et al., 2015 [44]	USA (U)	YO probiotic: 16 (not reported) YO placebo: 16 (not reported) YO total: (65–80)	YO probiotic: not reported YO placebo: not reportedYO total: 68.8	16S rRNA qPCR	α: No difference between placebo and probiotic groups (Chao1, observed OTUs) β: No difference between placebo and probiotic groups (UniFrac)	Genus: Probiotic ↑ *B**ifidobacteria* and ↑ lactic acid bacteria, compared to placebo Species: Probiotic ↓ *Escherichia coli* and ↑ *Faecalibacterium prausnitzii,* compared to placebo	N/A
Valentini et al., 2014 [45]	France, Germany, and Italy (U)	YO with diet and VSL#3 treatment: 31 YO with diet alone: 31 YO total (65–85)	YO total: 53.2 Not reported for each treatment arm	16S rDNA gene-targeted qPCR	α: N/A β: N/A	No change in *Clostridium* cluster IV, *Bifidobacterium* spp., after diet only and diet+VSL#3 treatment arms	N/A

Abbreviations used: ADT: adult; BCFA: branched chain fatty acids; CK: control group; CFA: cellular fatty acids; CHD: children; CTN: centenarian group; GF: gene family; KO: Kyoto Encyclopedia of Genes and Genomes (KEGG) orthology; LC-MS: liquid chromatography and mass spectrometry analysis; LL: long-living, oldest-old adults; N/A: not available or applicable; NHYO: non-healthy younger-old adult; OTU: operational taxonomic unit; PCR: polymerase chain reaction; PCR-DGGE: polymerase chain reaction- denaturing gradient gel electrophoresis; PD: phylogenetic diversity; R: rural; rDNA: ribosomal deoxyribonucleic acid; rRNA: ribosomal ribonucleic acid; SCTN: semi-supercentenarians; SCFA: short-chain fatty acid; U: urban; V: variable region of 16S rRNA; qPCR: quantitative polymerase chain; YO: young-old adult.

**Table 2 nutrients-12-03759-t002:** Summary of age-related findings for each gut microbiota metric across different categories of studies.

	Alpha Diversity	Beta Diversity	Taxonomic Differences	Functional Potential and Metabolites
(A) Long-Lived Individuals	5/5 studies 60% reported differences between LL younger age groups. 40% reported no differences across LL, YO, and ADT groups.	5/7 studies 100% reported differences between LL and younger age groups, including YO and ADT.	8/8 studies 37.5% reported Phylum level: LL ↑ Proteobacteria and ↓ Firmicutes 100% reported Family/Genus level: LL ↑ *Akkermansia*, *Christensenellaceae*, *Escherichia*, *Clostridium*, *Desulfovibrio*, *Parabacteroides, Odoribacter*, *Butyricimonas*, *Eggerthella*, and *Anaerotruncus*; ↓ *Faecalibacterium*, *Prevotella*, and *Bacteroides*.	4/4 studies 75% examined KEGG pathways; 25% studied gut metabolites.Beta diversity of KEGG gene pathways different between LL and YO and ADT. LL ↑ pathways related to central energy metabolism and respiration; ↓ pathways related to genetic information processing; ↓ pathways related to carbohydrate degradation and metabolism; ↓ vitamin B1 pathways, but ↑ B2 and K2 pathways; ↑ metabolite derivatives of butyrate.
(B) Lifespan	2/6 studies 50% reported ↑ alpha diversity with age (lowest at infancy, with increasingly higher levels through adolescence and young adulthood; stable across adult decades; higher in YO and LL). 50% reported no difference between YO and ADT.50% ↑ alpha diversity in community-dwelling YO compared to long-term care YO.	4/5 studies 25% reported that aging explained a significant proportion of variance in beta diversity distances across lifespan. 75% reported no difference between YO and ADT.33% reported differences between community-dwelling YO and long-term care YO.	9/10 studies 33% reported Phylum level: Actinobacteria highest in infants, lower after weaning and lower with age/development. YO ↑ Proteobacteria and Bacteroidetes.33% reported Family/Genus level: YO ↓ *Clostridium cluster XIVa* 70% reported Species level: *Bifidobacterium longum* present across the lifespan. *B. breve* most prevalent in infants, *B. adolescentis* in adults, and *B. dentium* in YO. YO ↑ specific *Lactobacillus* species (*L*. *paracasei*, *L. plantarum*, *L. salivarius*, and *L. delbrueckii*). YO ↓ *Faecalibacterium prausnitzii*.	7/7 studies 71.4% investigated metabolites; 14.2% studied gene families; 14.2% examined KEGG pathways. YO ↑ functional pathways related to drug transporters and ↑ gene clusters related to polysaccharide synthesis; ↓ gene families involved in genetic transcription, repair, and defense mechanisms, ↓ butyrate-producing gene (butyryl-CoA:acetate CoA-transferase). Aging associated with ↑ metabolism of aromatic amino acids and ↓ biosynthesis of amino acids, whereas adulthood associated with ↑ SCFA production. No difference in gut metabolites between YO and ADT.
(C) Cognition	0/1 study	0/0 studies	0/3 studies	0/0 studies
(D) Intervention	2/2 studies 100% reported no significant differences following probiotic or prebiotic in YO.	2/2 studies 50% studies performed placebo-controlled vs. treatment group comparisons; 50% examined pre- vs. and post-treatment comparisons.100% reported no significant differences following probiotic or prebiotic.	4/4 studies 75% of studies reported ↑ *Bifidobacterium*, 25% reported ↑ *Faecalibacterium prausnitzii*, 25% reported ↑ *Lactobacillus* spp., 25% reported ↑ *Lactobacillus acidophilus*, and 25% reported ↓ *Escherichia coli* following probiotic and synbiotic.No change in taxonomic composition following prebiotic.	1/1 study 100% found no significant differences in BCFA or SCFA following prebiotic.

Data reported as: number of studies that reported results related to age or aging/number of studies that reported any results on the specific metric. Abbreviations used: ADT: adult; BCFA: branched chain fatty acids; KEGG: Kyoto Encyclopedia of Genes and Genomes; LL: long-living, oldest-old adults; OTU: operational taxonomic unit; PD: phylogenetic diversity; SCFA: short-chain fatty acids; YO: young-old adults.

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
