# Peer review of "The Gut Microbiome, Aging, and Longevity: A Systematic Review"

_nutrients, 2020, doi:10.3390/nu12123759_

Round 1
Reviewer 1 Report
This review article is a well written topical systemic review and a valuable contribution the current literature. However, I believe that this article could benefit from addressing the points below:
1. The introduction is lacking clarity regarding the concept of immune senescence; it extends beyond a decline in the immune system. It is not clear why this has been mentioned in the introduction and abstract because the review has not looked for the potential impact on microbiome on immune ageing. However, it might be worth mentioning the gut -brain axis to provide background into why cognition has been assessed.
- The inclusion criteria for participants in the different studies is not clear. If the studies did not exclude for participants with co-morbidities, or for those who had recently suffered an infection or consumed antibiotics or other medications that can impact the microbiome this limitation needs to be acknowledged. This might be a factor that helps explain the heterogeneity in findings across different studies.
- The impact of host genetics and lifestyle factors such as diet on the microbiome have been discussed but no mention about other potential influencers such as physical activity levels. Also, there is no mention about a potential gender bias. Is it possible that gender differences in microbiome help explain why females live longer?
- In the interventions section it might also be worth looking at the effects of other dietary interventions such as the consumption of Mediterranean diet on the microbiome.
Overall, I greatly enjoyed the review and look forward to reading the final version.
Author Response
1. The introduction is lacking clarity regarding the concept of immune senescence; it extends beyond a decline in the immune system. It is not clear why this has been mentioned in the introduction and abstract because the review has not looked for the potential impact on microbiome on immune ageing. However, it might be worth mentioning the gut -brain axis to provide background into why cognition has been assessed.
Response: We thank the Reviewer for these suggestions. We have removed the term “immunosenescence” from the Abstract (line 20) and Introduction to avoid overemphasizing this point and confusion about the definition, and we have added sentences that more broadly connect immune system dysregulation with the gut-brain-axis (page 2, lines 53-62): “The microbiome is a principal factor in determining the immune system response and its dysregulation may sustain pro-inflammatory states [5]. The progression of aging involves a gradual weaking of the immune system resulting in an imbalance between pro-inflammatory and anti-inflammatory activity [6]. Age-related changes in pro-inflammatory status result in low-level systemic inflammation (“inflammaging”) that increases the propensity for chronic diseases and disabilities, including cardiovascular disease, cognitive decline, metabolic disease, frailty, and mortality [7,8]. Furthermore, gut microbes can communicate with the brain and modulate behavior, including higher-order cognitive functions, via the “gut-brain-axis” through neural, immune, and hormonal mediators [9]. Together, the microbiome offers an exciting perspective to understanding both physical and cognitive aspects of aging.”
2. The inclusion criteria for participants in the different studies is not clear. If the studies did not exclude for participants with co-morbidities, or for those who had recently suffered an infection or consumed antibiotics or other medications that can impact the microbiome this limitation needs to be acknowledged. This might be a factor that helps explain the heterogeneity in findings across different studies.
Response: The Reviewer raises a great point. We have added the following sentence in the Methods (page 3, lines 107-109): “While the inclusion criteria for participants across individual studies varied, a majority of investigations excluded subjects with major medical co-morbidities (18 out of 27) and recent antibiotic use (15 out of 27).”
We have also included the following statement in the Limitations (page 27, lines 374-378): “Additionally, while a majority of studies excluded subjects with major medical co-morbidities or recent antibiotic use, there was variability in inclusion criteria across studies with regard to the specific health status of participants, which may contribute to heterogeneity in findings across different studies.”
3. The impact of host genetics and lifestyle factors such as diet on the microbiome have been discussed but no mention about other potential influencers such as physical activity levels. Also, there is no mention about a potential gender bias. Is it possible that gender differences in microbiome help explain why females live longer?
Response: We thank the Reviewer for highlighting these points. We have added the following paragraph in the Discussion (page 27, lines 404-415): “A few studies acknowledged the possible role of sex [34] and decline in physical activity levels with age [28] on the microbiome and its impact on cognition [41]. Only one examined beta-diversity clustering by sex and found no differences in UniFrac distances between males and females from infants to centenarians [34]. While sex differences in the gut microbial composition have been documented [82], the influence of sex on the gut microbiome may be less influential than other clinical factors, such as genetics [83] or geographical origin [84]. As a result, it is unclear to what extent sex differences in the microbiome or physical activity might explain sex differences in longevity [85]. Physical activity is another important environmental factor that may influence the gut microbiome as well as aging and longevity. One study found that frailty, of which physical activity is a component, moderated the relationship between the microbiota and cognition [41], but no other article examined the role of exercise on the microbiome in aging, thus limiting conclusions that can be drawn.”
4. In the interventions section it might also be worth looking at the effects of other dietary interventions such as the consumption of Mediterranean diet on the microbiome.
Response: We appreciate the Reviewer’s suggestion. Of the studies that met inclusion criteria, only one investigated a dietary intervention with and without adjunctive probiotic treatment. We added results specific to the diet only treatment arm of this study in the Results section (page 24, lines 266-270): “With regards to dietary intervention, abundances of Clostridium cluster IV and Bifidobacterium were not altered following 8-weeks on the RISTOMED optimized diet intervention either alone or in combination with a probiotic supplement [45]. However, subgroup analysis revealed that individuals with low-grade inflammation showed an increase in Bifidobacterium following the dietary intervention with adjunctive probiotics.”
There were no articles on the Mediterranean diet that met the review criteria. We have added comment about findings of the Mediterranean diet on the gut microbiome in the Discussion (page 28, lines 428-440): “One study investigated the gut microbial effects of a dietary intervention with adjunctive probiotic treatment and found increased levels of Bifidobacterium among individuals with higher levels of systemic inflammation [45]. It is worth noting that the Mediterranean diet (e.g., high consumption of vegetables, legumes, fruits, nuts, olive oil, and fish; low consumption of red meat, dairy products, saturated fats, and processed foods) has been associated with improved health status, including reduced risk of mortality and occurrence of diseases of aging such as cardiovascular disease, cancer, and neurodegenerative disorders [87]. Prior studies have shown that intervention of and adherence to the Mediterranean diet is associated with lower Firmicutes–Bacteroidetes ratio, increased abundances of Christensenellaceae and Faecalibacterium prausnitzii, increased gene richness (particularly in those with low inflammatory status), and higher levels of gut SCFA in the general adult population [88,89]. Future studies should investigate the effects of the Mediterranean diet in older adults.”
Reviewer 2 Report
This paper takes high aim within a new field research, i.e. how the bacterial distribution/diversity may affect longevity. May be so, but the reviewer believes that the main point is what kind of metabolic products that are produced that will determine longevity. Take e.g. the production of colibactin by some E. coli that may have a direct DNA disrupting function which leds to malignant grows or possibly inflammation in the gut. The other part of interest is obesity where the bacterial flora has an important impact. What bacterial species are know to overload the human body with nutrients? Or possibly sugars to induce diabetes type 2. As longevity is usually dependent on the diseases an individual carries, one has to take the diseases into the concept.
Once we decide to study healthy individuals of high age without really considering diseases we cannot really speak of longevity, but rather healthy life unspecifically.
The authors have studied bacterial distributions which is a limit. We considered to be born aseptic but rapidly colonize our bodies from the mother's flora. So far it is OK, but after that it seems that the bacterial content of our gut (and possibly other places; toes, feet) increases. This has been described for the gut where old individuals are said to hold a greater bacterial flora and possibly higher bacterial content of the lower part of the small intestine with some tendency for small bowel bacterial overgrowth. The reviewer would wish for a better quantification of the bacterial counts within the gut over life time. If the bacterial flora has an evolving flora over life, what about the bacterial numbers?
I guess this is a limit to the present compilation of data. Distributions change quite clear but the total amount? And what should that mean?
The article is definitely worth publishing but some words on the bacterial numbers over a life span deserves some notification.
Author Response
Point 1. This paper takes high aim within a new field research, i.e. how the bacterial distribution/diversity may affect longevity. May be so, but the reviewer believes that the main point is what kind of metabolic products that are produced that will determine longevity. Take e.g. the production of colibactin by some E. coli that may have a direct DNA disrupting function which leads to malignant grows or possibly inflammation in the gut. The other part of interest is obesity where the bacterial flora has an important impact. What bacterial species are known to overload the human body with nutrients? Or possibly sugars to induce diabetes type 2. As longevity is usually dependent on the diseases an individual carries, one has to take the diseases into the concept.
Once we decide to study healthy individuals of high age without really considering diseases we cannot really speak of longevity, but rather healthy life unspecifically.
Response: We thank the Reviewer for raising this interesting and important point that longevity is dependent on diseases and can be understood in terms of diseases avoided in later years. We agree in that many extremely long-lived individuals, such as centenarians, may show a generalized decline in health and may exhibit some medical diseases, yet they manage to survive. For this reason, we did not exclude studies of individuals with co-morbid illnesses and posit that these findings are related to “normal” rather than “healthy” aging (see page 2, lines 64-77).
Additionally, in response to Reviewer 1’s comment (see response #2), we have added the following statement as a limitation of the review (page 27, lines 374-378): “Additionally, while a majority of studies excluded subjects with major medical co-morbidities or recent antibiotic use, there was variability in inclusion criteria across studies with regard to the specific health status of participants, which may contribute to heterogeneity in findings across different studies.”
As for how to address the issue of diseases among long-lived individuals, this is very difficult to address it from experimental design point of view and beyond the scope of this systematic review to answer.
Point 2. The authors have studied bacterial distributions which is a limit. We considered to be born aseptic but rapidly colonize our bodies from the mother's flora. So far it is OK, but after that it seems that the bacterial content of our gut (and possibly other places; toes, feet) increases. This has been described for the gut where old individuals are said to hold a greater bacterial flora and possibly higher bacterial content of the lower part of the small intestine with some tendency for small bowel bacterial overgrowth. The reviewer would wish for a better quantification of the bacterial counts within the gut over life time. If the bacterial flora has an evolving flora over life, what about the bacterial numbers?
I guess this is a limit to the present compilation of data. Distributions change quite clear but the total amount? And what should that mean?
The article is definitely worth publishing but some words on the bacterial numbers over a life span deserves some notification.
Response: This is again a very valid and important point raised by the Reviewer, i.e., requesting quantification of gut bacterial counts to aid in visualizing an overall “trajectory” across the lifetime. We agree that this is an important limitation to the current review due to the cross-sectional nature of the studies included, which makes it difficult to separate cohort effects from within-subject effects. We have included a statement of this limitation in the Discussion (page 27, lines 374-378): “Because of this, it is impossible to draw definitive conclusions about the longitudinal trajectories of bacterial counts or alpha-diversity across the lifespan.”
Furthermore, we believe that an absolute quantification of the bacterial numbers would carry little meaning because it would be confounded by many sample- and within-subject specific factors, e.g., amount of food intake before sampling, fasting, time spent in gut where nutrients are being reabsorbed and as a consequence, the microbiome readjusting to the dynamic conditions in the gut.